# Quality of life among patients with the common chronic disease during COVID-19 pandemic in Northwest Ethiopia: A structural equation modelling

Tadesse Awoke Ayele[1], Habtewold Shibru Fanta[2], Malede Mequanent Sisay[1], Tesfahun Melese Yilma[3], Melkitu Fentie[4], Telake Azale[5], Tariku Belachew[6], Kegnie Shitu [5], Tesfa Sewunet Alamneh [1] *

1 Epidemiology & Biostatistics Department, College of Medicine and Health Sciences, University of Gondar, Gondar, Ethiopia, 2 Internal Medicine Department, College of Medicine and Health Sciences, University of Gondar, Gondar, Ethiopia, 3 Health Informatics Department, College of Medicine and Health Sciences, University of Gondar, Gondar, Ethiopia, 4 Nutrition Department, College of Medicine and Health Sciences, University of Gondar, Gondar, Ethiopia, 5 Health Education & Behavioral Science Department, College of Medicine and Health Sciences, University of Gondar, Gondar, Ethiopia, 6 Amhara Health Bureau, Bahir-Dar, Ethiopia

* tesfasewunet23@gmail.com

**Data Availability Statement:** Data cannot be shared publicly because of the the ethical issue. Data are available from the university of Gondar

## Abstract

### Background

Improving Quality of Life (QoL) for patients with chronic diseases is a critical step in controlling disease progression and preventing complications. The COVID-19 pandemic has hampered chronic disease management, lowering patients' quality of life. Thus, we aimed to assess the quality of life and its determinants in patients with common chronic diseases, in Northwest Ethiopia during the COVID-19 pandemic.

### Methods

A cross-sectional study was conducted among 1815 randomly selected chronic patients with common chronic diseases. A standardized WHOQOL BREF tool was used, and electronic data collection was employed with the kobo toolbox data collection server. Overall QoL and the domains of Health-Related Quality of life (HRQoL) were determined. Structural equation modelling was done to estimate independent variables' direct and indirect effects. Path coefficients with a 95% confidence interval were reported.

### Results

About one in third, (33.35%) and 11.43% of the study participants had co-morbid conditions and identified complications, respectively. The mean score of QoL was 56.3 ranging from 14.59 and 98.95. The environmental domain was the most affected domain of HRQoL with a mean score of 52.18. Age, psychological, and environmental domains of HRQoL had a direct positive effect on the overall QoL while the physical and social relationships domains

Ethics Committee (contact via cmhsshirb2022@gmail.com) for researchers who meet the criteria for access to confidential data.

**Funding:** This study was funded by Ethiopian Ministry of Health with grant number of 34/49/1142. However, the funders had no role in study design, data collection and analysis, decision to publish, or preparation of the manuscript.

**Competing interests:** the authors have declared that no competing interests exist.

**Abbreviations:** CVD, Cardio-vascular disease; DM, Diabetes Mellitus; HRQoL, Health-Related Quality of life; IQR, Inter Quartile Range; QoL, Quality of Life; RMSA, Root Mean Error Approximation; SEM, Structural Equation Modelling; SDG, Sustainable Development Goal; WHO, World Health Organization's.

had an indirect positive effect. On the other hand, the number of medications taken, the presence of comorbidity, and complications had a direct negative impact on overall QoL. Furthermore, both rural residency and the presence of complications had an indirect negative effect on overall QoL via the mediator variables of environmental and physical health, respectively.

## Conclusion

The quality of life was compromised in chronic disease patients. During the COVID-19 pandemic, the environmental domain of HRQoL was the most affected. Several socio-demographic and clinical factors had an impact on QoL, either directly or indirectly. These findings highlighted the importance of paying special attention to rural residents, patients with complications, patients taking a higher number of medications, and patients with comorbidity.

## Background

The novel coronavirus Severe Acute Respiratory Syndrome (SARS-2 COVID-19) outbreak has resulted in a dramatic loss of human life and poses an unprecedented challenge to the world, particularly in developing countries [1]. The pandemic is continuing to cause over 3.5 million deaths in the world and aggravated hunger and poverty. Ethiopia had approximately 274,601 total cases and 4,260 deaths as of June 17, 2021 [2]. Following the confirmation of the first case, Ethiopia implemented various prevention strategies. These include closing schools and universities, avoiding large crowds, mandating mask wear, and restricting access to hospitals. Some patients were afraid to go to the hospital for treatment. COVID-19 has been linked to several adverse effects in patients with chronic diseases, including treatment delays, discontinuation, morbidity, and mortality [3–5].

However, chronic diseases such as HIV/AIDS, cardiovascular disease, cancer, diabetes, and chronic respiratory conditions (e.g., asthma and chronic obstructive pulmonary disease) continue to be the world's leading causes of death and disability [6–9]. About 41 million people died each year due to chronic illness, equivalent to 71% of all deaths globally, and over 85% of premature deaths occur in low- and middle-income countries [10]. These illnesses are associated with a decrease in patients' Quality of Life as well as significant socio-economic implications. Moreover, these conditions are associated with a decrease in patients' Quality of Life (QoL) as well as significant socio-economic implications [11].

The COVID-19 pandemic has had several impacts on the community psychosocial, health care system, and economy of the countries. Particularly for chronic disease patients, this pandemic increases the risk of severe illness and death [12, 13]. Control measures taken during the pandemic like the restriction of movement and mobilization of health professionals would also compromise chronic disease management and affects their QoL [14]. However, in patients with chronic disease, a complete cure is not possible; instead, supportive measures have been provided to control disease progression and prevent disease-related complications, with the ultimate goal of improving QoL [15, 16].

Despite COVID-19 having a tremendous effect on patients with chronic disease, there is limited evidence on QoL and its factors among chronic disease patients. Previous studies have been assessed using univariate analysis that only looks at the direct effect of the factors on the

dependent variable but not the indirect effect. While, QOL is a broad and multidimensional concept that includes domains related to physical, mental, emotional and social functioning [17].

Therefore, this study aimed at assessing the QoL of patients with common chronic diseases during the COVID-19 pandemic periods and its factors using Structural equation modelling (SEM). The study will provide evidence to policymakers and program planners to help them make decisions and will be useful for evidence-based interventions in support of the World Health Organization's (WHO) Sustainable Development Goal (SDG) target of a one-third reduction in premature deaths from the noncommunicable disease by 2030 [18].

## Methods

### Study design and setting

A cross-sectional study design was employed in public Hospitals that provide chronic care in Ethiopia, in 2021. The source population were all patients with common chronic diseases (HIV/AIDS, (Diabetes Mellitus) DM, cardiovascular disease (CVD), and respiratory diseases) that are on follow-up at the hospitals in Amhara regional state were the source population. Whereas patients who had chronic care appointments and follow up during the data collection period were the study population. All patients with common chronic disease conditions, aged at least 18 years who have been on medication for greater than or equal to 2 years were included. Those who repeated their visit during the data collection period were excluded from the study. The study included all hospitals (referral and district) with chronic care centres in the Amhara region.

### Source and study population

The source population included all patients with common chronic diseases (HIV/AIDS, DM, CVD, and respiratory diseases) who were being followed up on at hospitals in Amhara regional state. The study population consisted of patients who had chronic care appointments and follow-ups during the data collection period. All patients with common chronic disease conditions who were at least 18 years old and had been on medication for at least 2 years were included. Those who revisited during the data collection period were not included in the study.

### Sample size and sampling procedures

Regarding sample size estimation, there is no single formula used to estimate sample size for studies based on the Structural Equation Modelling (SEM) analysis approach. Several factors are considered during a sample size calculation, including the indicator-to-latent variable ratio, model complexity, assumption violations (i.e. multivariate normal distribution), and indicator reliability [19]. In point of this, the minimum required sample size for SEM is to be at least 200, 10 observations per observed variable and 20 cases per parameter estimated but none of this can't fit all in a different situation [10, 19, 20]. As a result, we decided to use a formula that can provide a maximum sample size to compute estimates even when the data do not conform to the multivariate normality assumption. Thus, $n = \frac{p(p+1)}{2}$, where p is 45 (the number of observed variables in the model). Accordingly, the sample size was 1355 including a 10% non-response rate and a design effect of 2.

In the Amhara region, study samples were drawn from both referral and district hospitals. First, stratification was performed based on hospital status (referral vs district). The hospitals were chosen at random. Finally, participants in the study were chosen using a systematic random sampling technique with disease type (HIV/AIDS, DM, CVD, respiratory illnesses, and cancer) in the selected hospitals while taking the proportion of disease categories into account (Fig 1).

## Data collection and measurements

The data were collected by using an interviewer-administered semi-structured questionnaire adapted from the WHO Quality of Life-BREF (WHOQOL-BREF) tool and developed after a review of various works of literature. Following training, health professionals and medical doctors with experience in chronic disease follow-up participated in data collection. An electronic questionnaire form was developed by the Kobo toolbox. The tool's validity and reliability were checked after the pretest. Possible changes to the data collection tool were made. The collected data was centrally reviewed daily for completeness and consistency. Following the retrieval of the appointment logbook and patient chart, patients were interviewed.

The instrument was divided into four sections: socio-demographic variables (age, gender, residence, marital status, literacy status), clinical factors (duration of follow-up, frequency of follow-up, number of medications, mental health problems, presence of co-morbidity and complication), behavioural factors (alcohol use and medication adherence), and the WHOQOL-BREF tool. The WHOQOL-BREF consists of 26 items designed to assess four domains of HRQoL as well as the overall perception of quality of life and general health. These are the Physical Health Domain (PHD) (7 items), the Psychological Health Domain (PSHD) (6 items), the Social Relationship Domain (SRD) (3 items), and the Environmental Health Domain (EHD) (8 items), the overall perception of QoL (2 items). Each item on the WHOQOL-BREF is assigned a score from 1 to 5, based on a five-point Likert scale [21]. To make the domains comparable, the domain raw score was calculated by multiplying the mean score of all items in each domain by 4, which ranged from 4 to 20 points. Then the raw score was linearly transformed to domain scores out of 100 by the formula; $domain\ score\ =\ (raw\ score - 4) * \frac{100}{16}$. Finally, the overall

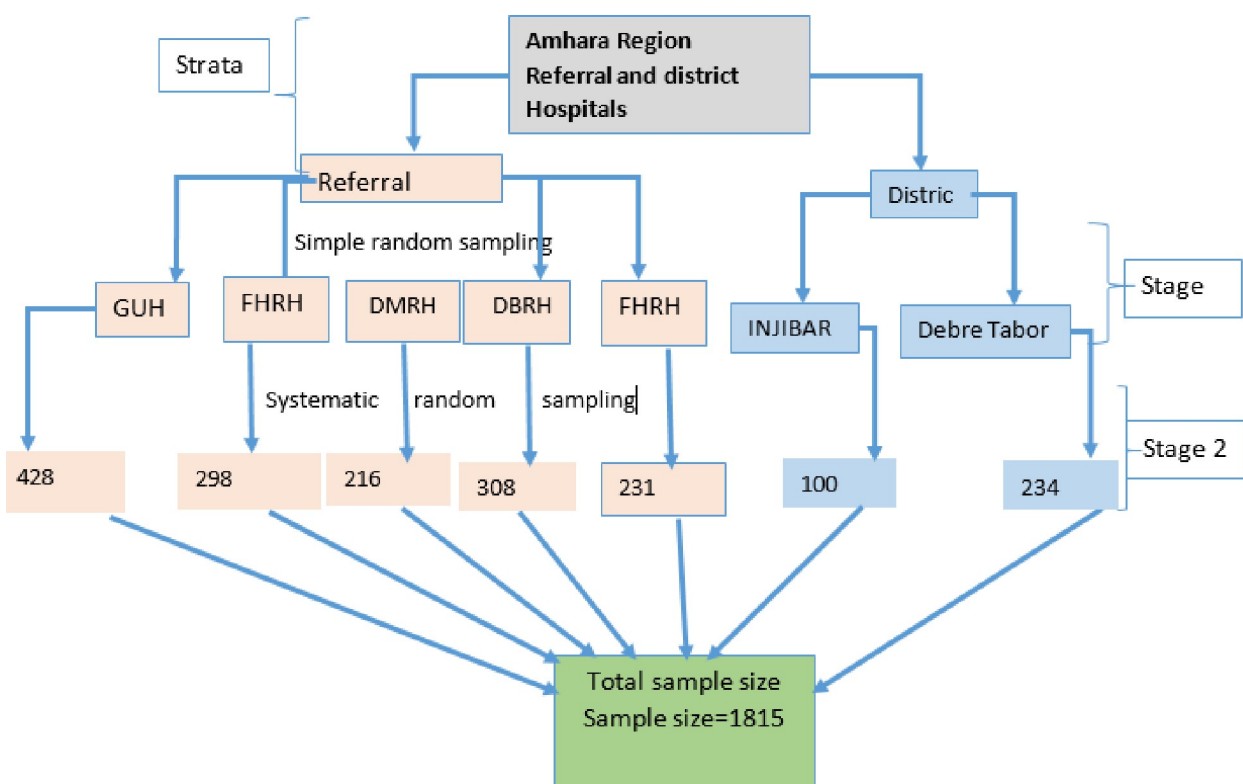

**Fig 1. Schematic presentation of sampling procedures (note that; RH and referral hospital, DH: District hospital).**

QoL was computed by the average of the four domain scores [22, 23]. Furthermore, the presence of common mental health problems was assessed using the SRQ-20 tool that is validated in Ethiopia with high internal consistency, Cronbach's alpha = 0.92 [24, 25]. To assess medication adherence, the Morisky Medication Adherence Scale (MMAS-8) was used, which consists of eight yes/no questions with a score ranging from 0 to 8. The internal consistency of the MMAS was checked out by Cronbach's alpha and was 0.72 [26, 27].

## Operational definition

**Chronic diseases.** Chronic diseases are broadly defined as conditions that last 1 year or more and require ongoing medical attention or limited activities of daily living or both [9]. In our setup, major chronic diseases under organized follow-up are cardiovascular diseases (hypertension, Chronic Kidney Disease (CKD), and cardiac illnesses), cancer, Respiratory disorder (COPD and Asthma), diabetes, chronic liver diseases, and HIV/AIDS [28].

## Data management and statistical analysis

All data were analyzed using R. For numerical variables, descriptive statistics such as mean, median, interquartile range, and standard deviation was used. For categorical variables, frequencies and percentages were used.

Following descriptive data exploration, SEM was used to assess the direct and indirect effects of factors on patients' overall QoL. The SEM was made up of two parts: the measurement model and the structural model. The measurement components evaluate the relationship between a latent variable and its indicators or items, whereas the structural components primarily indicate the relationship between the latent variables. It also provided causality between the system's dependent and independent variables [29]. The analysis began with the theoretical model (Fig 2) [30], and iterative modifications were made by adding paths or including mediator variables, if theoretically supported, and comparing by Root Mean Error Approximation (RMSA) as the absolute measure of the model fitness index and by information criteria as the measure of model parsimony. Finally, an overidentified model with an RMSA close to 0.05 and the smallest information criterion was kept. The effect of each exogenous or mediating variable on the respective dependent variable was represented diagrammatically by the path coefficient and a single-headed arrow, and the correlation among error terms was represented by double-headed arrows. To determine statistical significance, a confidence level of 95% and a p-value less than 0.05 were used.

## Ethical consideration

Ethical clearance was secured from the institutional review board of the University of Gondar. The supportive letter was obtained from the Amhara public health institute and permission was obtained from the medical director of each hospital. Participants of the study were informed about the purpose, objectives, and their right to participate or not participate in the research. The right of participants to withdraw from the study at any time, without any precondition was disclosed unequivocally. Written consent was obtained from each participant before data collection. Moreover, to guarantee confidentiality code numbers were used rather than personal identifiers.

## Results

### Background characteristics

Out of 1815 patients, more than half (55.37%) were female. Their median age was 48 years with an Inter Quartile Range (IQR) of 22 (37–59) years. Most of the study participants, 1,262

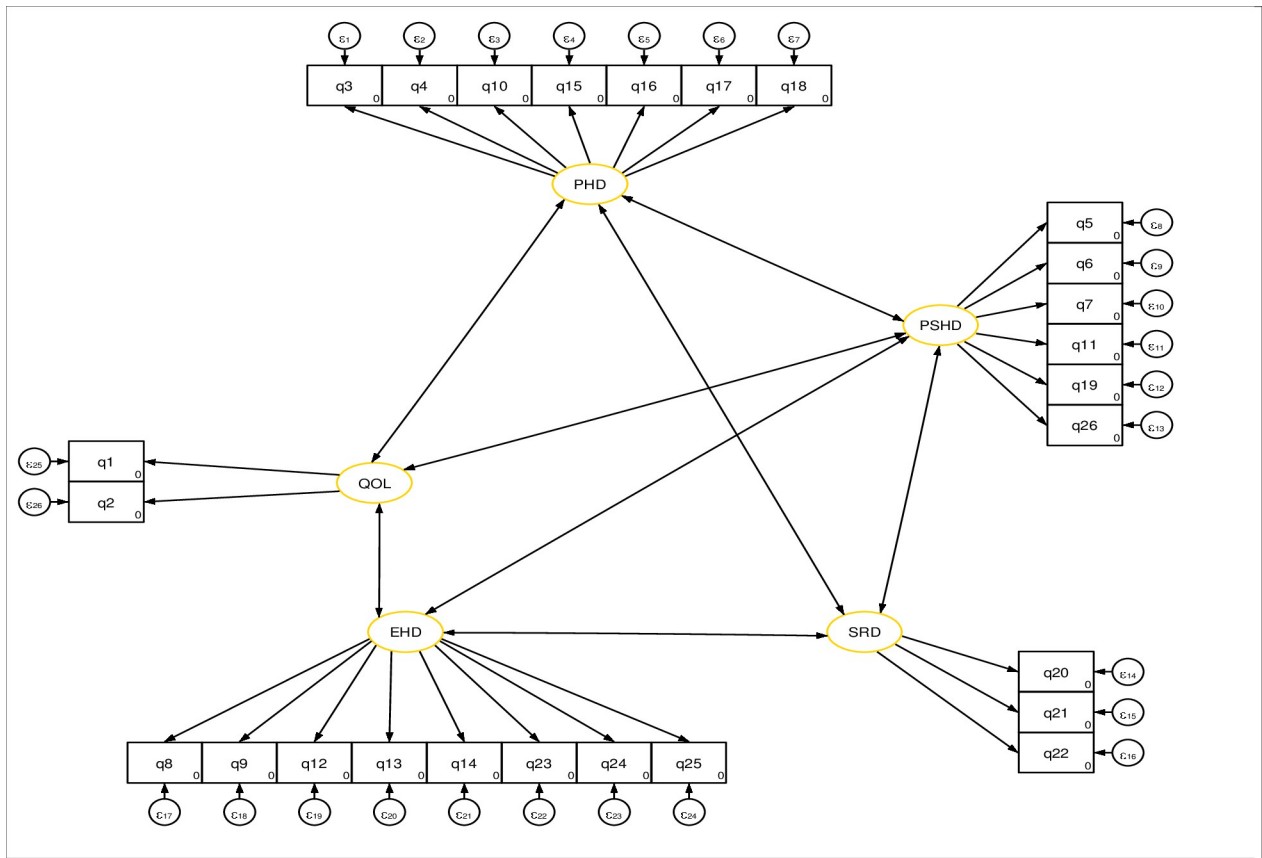

**Fig 2. Theoretical model of quality-of-life measurement (S1 Table).**

(69.53%) were urban dwellers. More than one-fourth (28.0%) of them were not able to read and write. Nearly one-in-four patients (23.63%) have been following for more than 10 years with a median duration of 6 years (IQR = 3–10 years). About one in third, 602 (33.35%) of the patients had identified co-morbid conditions and 205 (11.43%) of the study participants had identified complications (Table 1).

Regarding the chronic disease category, the three common types of chronic disease included in the study were HIV/AIDS (27.69%), hypertension (23.41%) and diabetics (22.7%) (Fig 3).

## Self-rated quality of life and perceived health satisfaction

More than half (52.62%) of the patients reported that their quality of life was good. Regarding perceived satisfaction with their health, 52.23% (948) were satisfied. On the other hand, one in four patients (26.23%) was not satisfied with their health (Table 2).

## Quality of life using the WHOQOL-BREF tool

The overall QoL mean score was 56.3 (±14.5). In terms of overall QoL, approximately 906 (49.92%) of the respondents scored below the mean, with minimum and maximum mean scores of 14.59 and 98.95, respectively. Cronbach alpha was used to assess the internal reliability of each domain, and all domains achieved good reliability (α ≥ 0.7). Respondents scored

**Table 1. Background characteristics of patients with common chronic disease in Amhara region, Ethiopia, 2021.**

| Variables | Category | Number | % |
|---|---|---|---|
| Age | Median ± IQR | 48 ± 22 | |
| Sex | Female | 1,005 | 55.37 |
| | Male | 810 | 44.63 |
| Residence | Urban | 1,262 | 69.53 |
| | Rural | 553 | 30.47 |
| Marital status | Single | 218 | 12.01 |
| | Married | 1,187 | 65.40 |
| | Divorced | 107 | 5.90 |
| | Separated | 115 | 6.34 |
| | Widowed | 188 | 10.36 |
| Literacy status | Unable to read and write | 508 | 27.99 |
| | Able to read and write | 1123 | 72.01 |
| Education status | Primary education | 326 | 39.28 |
| | Secondary education | 298 | 35.90 |
| | Diploma | 206 | 24.82 |
| Medication adherence | Low | 310 | 17.08 |
| | medium | 977 | 53.83 |
| | high | 528 | 29.09 |
| Alcohol use | No | 1,550 | 85.40 |
| | Yes | 265 | 14.60 |
| Duration of follow up | < 2 years | 373 | 20.45 |
| | 2–5 years | 534 | 29.28 |
| | 6–10 years | 486 | 26.64 |
| | >10 years | 431 | 23.63 |
| Frequency of follow up | Weekly | 25 | 1.37 |
| | Every two/three week | 91 | 5.00 |
| | Monthly | 554 | 30.42 |
| | Every two month | 402 | 22.08 |
| | Every 3–5 months | 585 | 32.13 |
| | Every 6 or more months | 164 | 9.01 |
| Presence of co-morbidity | No | 1,203 | 66.65 |
| | Yes | 602 | 33.35 |
| Presence of complication | No | 1,588 | 88.57 |
| | Yes | 205 | 11.43 |

the highest in the social domain (62.93 ± 15.43) of the four HRQoL domains. Patients, on the other hand, scored the lowest in the environmental domain (52.18 ± 14.96) (Table 3).

## Causal factors for quality of life among chronic disease patients

Fig 4 depicts the final model, which contains both the measurement and structural components of structural equation modelling. When compared to other fitted models, these models had an RMSA of 0.08 and lower Akaike information criteria and Bayesian information criterion values, so they were chosen as the relatively fitted model. Some variables, such as marital status, education, current alcohol use, and follow-up duration, were excluded from the final model because their estimated contributions were not statistically significant at an alpha level of 0.05.

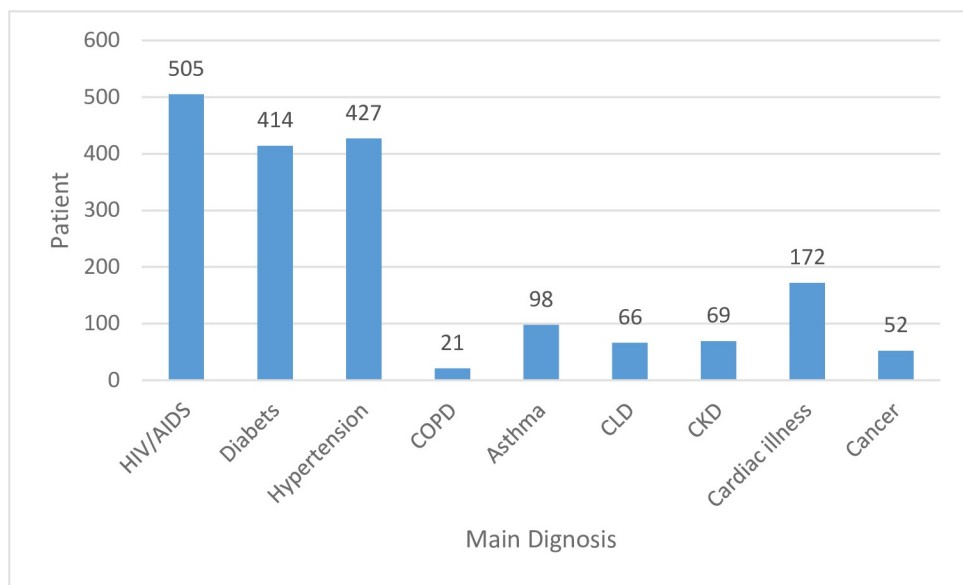

**Fig 3. Type of chronic disease in patients on chronic disease follow-up at Amhara region, Ethiopia, 2021.**

In the final model, all the path coefficients in the diagram were statistically significant at an alpha level of 0.05. Accordingly, this model included 8 exogenous observed variables (age, sex, residence, medication adherence, kinds of medication in numbers, presence of comorbidity, presence of complications, and mental health problem), four mediator latent variables (the four domains of QoL), one endogenous latent variable (QoL), and 26 endogenous observed variables (items of QoL). The exogenous observed variables, namely age, sex, residence, medication, kinds of medication, presence of comorbidity, mental health problem, and presence of complications were, directly and indirectly, related to QoL through the mediator variables of physical health domain, psychological health domain, social relationship domain and environmental health domain.

Specifically, age (adjusted β = 0.054, 95%CI;0.026, 0.082), psychological (adjusted β = 0.80, 95%CI;0.62, 0.98), environmental (adjusted β = 0.23, 95%CI;0.06, 0.39) had a direct positive effect on QoL. Besides, the physical and social relation domain (adjusted β = 0.184, 95%CI, 0.051, 0.339) had a positive indirect effect on QoL. However, kinds of medication (adjusted β = -0.03, 95%CI;-0.060, -0.004), presence of comorbidity (adjusted β = -0.033, 95%CI; -0.062, -0.002), and presence of complication (adjusted β = -0.023,95%CI;-0.056, -0.003) had a direct negative effect on QoL. Moreover, rural residency (adjusted β = -0.006, 95%CI;-0.016,-0.0005) and the presence of complication (adjusted β = - 0.045, 95%CI;-0.069, -0.024) had an indirect negative effect on the QoL via the mediator variable environmental and physical health domains respectively. In addition, the psychological (adjusted β = 0.80, 95%CI; 0.62, 0.98) and

**Table 2. Self-rated quality of life and health status satisfaction among chronic disease patients in Amhara region, Ethiopia (n = 1, 815).**

| Self-rated quality of life | | Satisfaction with health status | |
|---|---|---|---|
| Response category | Frequency, n (%) | Response category | Frequency, n (%) |
| Very poor | 32 (1.76) | Very dissatisfied | 31 (1.71) |
| Poor | 445 (24.52) | Dissatisfied | 445 (24.52) |
| Neutral | 383 (21.10) | Neutral | 391 (21.54) |
| Good | 857 (47.22) | Satisfied | 847 (46.67) |
| Very good | 98 (5.40) | Very satisfied | 101 (5.56) |

**Table 3. Quality of life descriptive results among chronic disease patients in Amhara region, Ethiopia (n = 1, 815), 2021.**

| Domains | Cronbach's α | Median | Mean ± SD | 95% CI |
|---|---|---|---|---|
| Physical | 0.85 | 57.14 | 55.89 ± 18.09 | (55.06, 56.73) |
| Psychological | 0.84 | 58.33 | 58.78 ± 17.46 | (57.98, 59.59) |
| Social relationships | 0.70 | 66.67 | 62.93 ± 15.43 | (62.22, 63.64) |
| Environmental | 0.81 | 53.13 | 52.18 ± 14.96 | (51.49, 52.88) |
| Overall QoL | - | 57.29 | 56.26 ± 14.51 | (55.59,56.93) |

environmental (adjusted β = 0.23, 95%CI; 0.06, 0.39) domains of health had a direct positive effect where as the physical (adjusted β = 0.69, 95%CI; 0.43, 1.02) and social (adjusted β = 0.18, 95%CI; (0.05, 0.34) domains of health had an indirect positive effect on the QoL (Table 4).

## Discussion

This study aimed to assess the quality of life with its domains and casual factors among patients with the common chronic disease during the COVID-19 pandemic period. This study estimates the mean score of the domains of HRQoL and overall QoL for patients with common

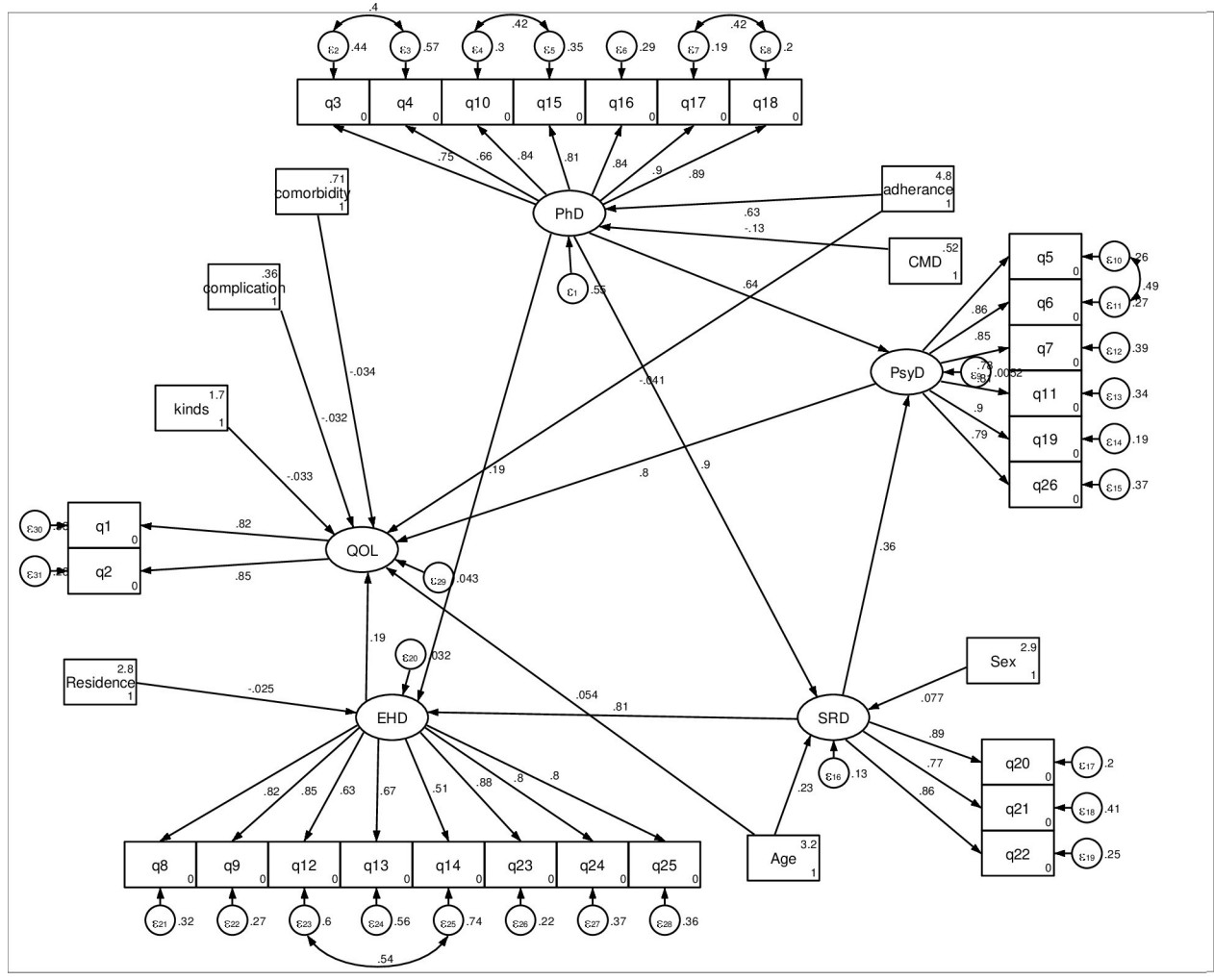

**Fig 4. Causal factors for quality of life among common chronic disease patients in Amhara region, Ethiopia, derived from the SEM, 2021.**

**Table 4. The direct, indirect, and total effects of sociodemographic and clinical factors on quality of life among common chronic disease patients in Amhara region, Ethiopia, derived from the SEM, 2021.**

| Characteristics | Category | Direct effect (95%CI) | Indirect effect (95%CI) | Total effect (95%CI) |
|---|---|---|---|---|
| **DV: QoL** | | | | |
| Physical domain | | - | 0.69 (0.43, 1.02) | |
| Psychological domain | | 0.80 (0.62, 0.98) | - | |
| Social relationship domain | | - | 0.184 (0.051, 0.339) | |
| Environmental domain | | 0.23 (0.06, 0.39) | - | |
| Age | | 0.054 (0.026,0.082) | - | |
| Residence | Urban | 0 | 0 | |
| | Rural | - | -0.006 (-0.016, -0.0005) | |
| Mental health problem | No | 0 | 0 | |
| | Yes | - | -0.045 (-0.069, -0.024) | |
| Presence of complication | No | 0 | 0 | |
| | Yes | -0.023 (-0.056, -0.003) | - | |
| Presence of comorbidity | No | 0 | 0 | |
| | Yes | -0.033(-0.062, -0.002) | - | |
| Kinds of medication (number) | | -0.03(-0.060, -0.004) | - | |
| **DV*: physical health domain** | | | | |
| Medication adherence | | 0.63 (0.61, 0.64) | - | |
| Mental health problem | No | 0 | 0 | |
| | Yes | -0.13 (-0.17, -0.09) | - | |
| **DV: Psychological domain** | | | | |
| Age | | - | 0.083 (0.065, 0.103) | |
| Sex | Female | 0 | 0 | |
| | Male | - | 0.028 (0.016, 0.041) | |
| Mental health problem | No | 0 | 0 | |
| | Yes | -0.051(-0.069, -0.033) | -0.098 (-0.13, - 0.071) | -0.149 (-0.199, -0.104) |
| Social relationship domain | | 0.35 (0.29, 0.41) | - | |
| Physical domain | | 0.65 (0.59, 0.71) | - | |
| **DV: Social relation domain** | | | | |
| Age | | 0.21 (0.19, 0.24) | - | |
| Sex | Female | 0 | 0 | |
| | Male | 0.077 (0.053, 0.10) | - | |
| Physical domain | | 0.90 (0.89, 0.92 | - | |
| Medication adherence | | - | 0.57 (0.54, 0.59 | |
| Mental health problem | No | 0 | 0 | |
| | Yes | - | -0.082 (-0.11, -0.06 | |
| **DV: Environmental domain** | | | | |
| Residence | Urban | 0 | 0 | |
| | Rural | -0.016 (—0.033, -0.001) | - | |
| Social relation domain | | 0.80 (0.74,0.87) | - | |
| Physical domain | | 0.19 (0.13, 0.26) | 0.73 (0.66,0.80) | 0.92 (0.73, 1.06) |
| Age | | | 0.19 (0.14, 0.21) | |
| Sex | Female | 0 | 0 | |
| | Male | | 0.06 (0.04, 0.09) | |
| Mental health problem | No | 0 | 0 | |
| | Yes | - | -0.025 (-0.044, -0.011) | |
| Medication adherence | | - | 0.12 (0.08, 0.17) | |

*DV = Dependent Variable

chronic diseases during the COVID-19 pandemic. The study found that patients with common chronic diseases had compromised QoL in all domains, especially in the environmental health domain. The finding was in agreement with previous studies done in Ethiopia among diabetic and heart failure patients [23, 31]. This could be explained by the COVID-19 pandemic creating enormous direct or indirect economic, financial, psychological, and institutional burdens to the society next to the second world war [32, 33]. The control measures for COVID-19 infection like moment restriction or lockdown and social distancing also highly compromise environmental health factors like physical security, financial resources and healthcare facilities [34]. Moreover, it led to the generation of a massive amount of medical waste that affects the air condition and water quality across the globe which in turn affects the QoL [35].

This study also identifies the associated factors of QoL. The study revealed that being a rural resident was associated with lower HRQoL in a social relation domain and overall QoL as compared with their counterparts. This finding was in agreement with a study done at Ethiopia [36]. This could be justified by the fact that rural dwellers might have poor social relationships, support and limited availability and accessibility of health facilities that play a great role in chronic disease follow-up and management [37]. Thus, individuals with limited accessibility and availability of health facilities had poor disease control and management which in turn compromised the quality of life [38].

Consistent with a study done in Egypt [22], this study showed that the age of the patients had a direct positive relationship with QoL. This could be linked with as age increases the patients might hold fewer responsibilities to think about with regard to their work and their families as compared with younger patients [39]. In addition, as age increases patients might build good social relationships and gain better social support from society and their families [40].

This study highlighted that the presence of comorbidity had an inverse relationship with QoL. The result of this study agreed with previous studies conducted in Ethiopia, Iran, and Malaysia [23, 41, 42]. The possible justification for this finding could be the presence of comorbidity makes the patient become dependent on many different drugs. Thus, this patient needs extra money to afford these drugs and the demand for healthcare services [43]. On the other side, taking many medications contributes to impaired QoL due to their side effects or drug interactions of the different drugs [44].

In addition to comorbidity, the presence of an identified complication/complications also inverse relationship with QoL, which was supported by previous studies done in Kenya, Saudi Arabia, and the United States of America that show the presence of an identified complications was negatively associated QoL [23, 45, 46]. This might be explained by the presence of complications is an indicator of poor treatment control and disease follow-up [47]. While disease control and strict follow are recommended in patients with chronic disease with the goal of enhancing QoL.

Medication adherence had a positive relationship with the Physical domain of WHO BREF while the number of medications taken for controlling the disease had an inverse relation with the overall QoL [48, 49]. This could be justified by patients taking the medication adherently may be associated with relieving signs and symptoms of their underlying disease in the short term and helps to manage their underlying disease condition in the long term period, thus resulting in better social, physical functioning and improved their QoL [50, 51].

The current study used a standardized tool, and data were collected by trained and experienced nurses and medical doctors under close and supportive supervision. The respondents were also informed about the importance of the study and the confidentiality of personal data to gain the trust of respondents and minimize the nonresponse rate. But this study was not free of limitations. The study includes different medical conditions; thus, the quality of life and

its factors might be different for each disease entity attention should be given while interpreting the findings of the study. Moreover, since the study was facility based there might be a risk of social desirability bias. Also, the application of SEM for latent variables like quality of life is considered the main strength of the study but the model only accommodate binary variables as in the measurement component which might cause potential loss of information.

## Conclusions

The quality of life of patients with common chronic diseases was compromised during the COVID-19 pandemic. The environmental domain of health was the most affected domain of health-related quality of life. The socio-demographic variables (age, sex, and residence), clinical factors (kinds of medications, presence of co-morbidity, complication, and mental health problem), and medication adherence from the behavioural factors had either direct or indirect significant relation with QoL. Therefore, program planners and policymakers should give special emphasis to rural residents, patients with complications, taking higher numbers of medications, and co-morbidity. Further, improving the patient's behaviour on medication adherence had a para-amount importance for enhancing their QoL.

## Supporting information

**S1 Table. WHOQOL BRIEF tool description.**
(DOCX)

## Acknowledgments

The authors are grateful to the federal ministry of health (MoH) for sponsoring this research. We would like to thank the University of Gondar, Amhara Health Bureau and Amhara Public Health Institute for the technical support and facilitation they provide during the study. We also thank the study participants for providing the information during the interview.

## Author Contributions

**Conceptualization:** Tadesse Awoke Ayele, Malede Mequanent Sisay, Tariku Belachew.

**Formal analysis:** Tadesse Awoke Ayele, Habtewold Shibru Fanta, Tesfa Sewunet Alamneh.

**Funding acquisition:** Tadesse Awoke Ayele.

**Investigation:** Tesfahun Melese Yilma, Melkitu Fentie, Kegnie Shitu.

**Methodology:** Habtewold Shibru Fanta, Malede Mequanent Sisay, Tesfahun Melese Yilma, Melkitu Fentie, Telake Azale, Kegnie Shitu, Tesfa Sewunet Alamneh.

**Software:** Tadesse Awoke Ayele, Tesfa Sewunet Alamneh.

**Supervision:** Tadesse Awoke Ayele, Malede Mequanent Sisay, Tesfahun Melese Yilma, Melkitu Fentie, Telake Azale, Tariku Belachew, Kegnie Shitu, Tesfa Sewunet Alamneh.

**Validation:** Habtewold Shibru Fanta, Malede Mequanent Sisay, Tesfahun Melese Yilma, Melkitu Fentie, Telake Azale, Tariku Belachew, Kegnie Shitu.

**Visualization:** Habtewold Shibru Fanta, Telake Azale, Kegnie Shitu.

**Writing – original draft:** Tesfahun Melese Yilma, Melkitu Fentie, Tesfa Sewunet Alamneh.

**Writing – review & editing:** Tadesse Awoke Ayele, Habtewold Shibru Fanta, Malede Mequanent Sisay, Tesfahun Melese Yilma, Melkitu Fentie, Telake Azale, Tariku Belachew, Kegnie Shitu, Tesfa Sewunet Alamneh.

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
