## [Decision Letter · Decision Letter 0]

26 Apr 2022

PONE-D-21-23936Quality of life among patients with common chronic disease during COVID-19 Pandemic in Northwest Ethiopia; A Structural Equation ModelingPLOS ONE

Dear Dr. Alamneh,

Thank you for submitting your manuscript to PLOS ONE. After careful consideration, we feel that it has merit but does not fully meet PLOS ONE’s publication criteria as it currently stands. Therefore, we invite you to submit a revised version of the manuscript that addresses the points raised during the review process.

We look forward to receiving your revised manuscript.

Kind regards,

Filipe Prazeres, MD, MSc, Ph.D.

Academic Editor

PLOS ONE

Journal Requirements:

Reviewers' comments:

Reviewer's Responses to Questions

**Comments to the Author**

1. Is the manuscript technically sound, and do the data support the conclusions?

Reviewer #1: Partly

Reviewer #2: Yes

Reviewer #3: Partly

2. Has the statistical analysis been performed appropriately and rigorously? 

Reviewer #1: Yes

Reviewer #2: Yes

Reviewer #3: Yes

3. Have the authors made all data underlying the findings in their manuscript fully available?

Reviewer #1: Yes

Reviewer #2: No

Reviewer #3: Yes

4. Is the manuscript presented in an intelligible fashion and written in standard English?

Reviewer #1: No

Reviewer #2: No

Reviewer #3: No

5. Review Comments to the Author

Reviewer #1: This is a cross sectional study to understand the determinants that affects QOL in patients with chronic diseases during COVID pandemic in Ethiopia. My comments are as follows:

The study design:

Although the authors mentioned in both discussion and conclusion that the quality of life was compromised during COVID-19, this conclusion cannot be drawn from the existing data. This is a cross sectional study - unless the current data can be compared to similar data prior to the COVID-19 outbreak, we cannot know if the QOL actually worsened during COVID-19.

Furthermore, the determinants that may predict QOL appeared to be similar to pre-COVID19 era. i.e. poorer QOL is found in people with disease complications. What's the new information then?

Results:

1/ The age of the participants are relatively young (they were all supposed to have chronic diseases). Why is that so? Most of the patients with chronic diseases are usually eldery. (your IQR was 37-59).

2/ some of the results are difficult to read. for example, "age.... had direct posetive effect on QOL". Does this mean younger or older age? The same goes with the other presentations.

3/ "kinds of medication.... had direct negative effective on QOL". How do you put different kinds of medications into a statistical model? Furthermore, what are the medications that are associated with low QOL?

4/ I will be very careful to use the word "Causal factors" because this is a cross-sectional study, many causal relationships cannot be established. For example, it is likely that people with good drug compliance has less problems with their diseases and thereby a better QOL. However, it may also be true that people with good QOL are less troubled by their problems and thereby remember to take their medications. Actually this is one of the limitations that should be discussed.

Discussion:

1/ most of the discussion were concentrated to explain the results. But some of the results may not need very detailed discussion. For example, it is well understood why presence of co-morbidities is linked to poorer QOL. There should be multiple reasons behind. Rather, the discussion can point out how we can use the results. Any policies that need to be changed according to your results? Any more research that needs to be done?

2/ The authors correctly pointed out that the study population is heterogeneous - it included multiple chronic diseases. Then, the authors can see if the results are the same when analyzing only a subgroup of patients (e.g. only include patients with HIV or patients with hypertension). This will make your results more robust

3/ The findings may actually apply to pre-COVID era. Can the data be compared to other studies in the pre-COVID era to show that there is indeed a worsening of environmental QOL, for example?

Minor:

There are multiple spelling and grammar mistakes. The manuscript may benefit from language editing.

Reviewer #2: /// Overall ///

I don't have background on direct and indirect effects, the authors may want to clarify these to all readers. Overall, I understand the the direct effect was determined via univariate model(s) whilst the indirect one was from a mixed multivariate model(s), then, why you have to call it direct and indirect?

Typo are found in the text, please proofread carefully

//// Detail suggestions////

line 32: the range of mean score QOL seem to be large, author may consider to report both mean and median

line 161: the full term of CKD should be written before the abbreviation. Please check through all abbreviations.

line 171, 231: typo

line 205: reference of reliability thresholds should be cited properly.

line 262-266: also many literature have proved that QOL decrease by age (as their physicals health getting poorer), you may want to add some discussion on this phenomenon.

/// Tables and Figures ///

Figure 2 and 4: the author may want to add some notes to explain what are q1-q25, PHD, QOL, EHD, SRD, PSHD

Figure 3: typo

Reviewer #3: Abstract and Introduction- Need to rephrase and construct for better sentences

Line 54-58 Need to check the sentences, the facts are contradicting with each other.

Line 96- suggest for ‘Facility based cross sectional study design’ to ‘A cross sectional study design’…

Methods- study setting

It is important to highlight during the study period, how many existing healthcare facilities are used for COVID-19, converted as hybrid healthcare facilities and healthcare facilities are not used for COVID-19. Not need to mention too detail about the burden of disease for COVID-19.

Can you explain about the semi-structured questionnaire used for your study?

What is the proportion of face-to-face interview before you switch to the electronic form?

Why were the validity and reliability not conducted for SRQ-20 tool and Morisky Medication Adherence Scale (MMAS-8)?

The chronic diseases are self-reported or confirmed with the medical records?

There are few typos error

How do you overcome information and recall bias in this study?

Should separate the categories able to read and write and the education level and re-analyze the data.

Can the patient the differentiate the symptoms of the disease and complications of the disease?

How did you ask the common mental disorders for this study? (This is a diagnosis)

What was measured for ‘Kind of medication’?

6. PLOS authors have the option to publish the peer review history of their article (what does this mean?). If published, this will include your full peer review and any attached files.

Reviewer #1: No

Reviewer #2: No

Reviewer #3: No

---

## [Author Response · Author response to Decision Letter 0]

4 Jul 2022

June 2022

Rebuttal letter

Manuscript ID: PONE-D-21-23936

Title: Quality of life among patients with common chronic disease during COVID-19 Pandemic in Northwest Ethiopia; A Structural Equation Modelling

Tadesse Awoke Ayele, HabtewoldShibru, Malede Mequanent Sisay, TesfahunMelese, MelkituFentie, TelakeAzale, Tariku Belachew, Kegnie Shitu, and Tesfa SewunetAlamneh*

PLOS ONE

Dear Editor and reviewer, 

We would like to thank for your consideration and suggestion for the betterment our manuscript and make it more informative. We tried to amend the format of the manuscript according to the journal guidelines and address the questions raised by reviewer on the manuscript. Our point-by-point responses for each comment and questions are described in detail on the following pages. Further, the details of changes were shown by track changes in the supplementary document attached. 

Editor’s comment

https://journals.plos.org/plosone/s/file?id=wjVg/PLOSOne_formatting_sample_main_body.pdf andhttps://journals.plos.org/plosone/s/file?id=ba62/PLOSOne_formatting_sample_title_authors_affiliations.pdf

Authors’ response: Thank you dear editors for your concern. We tried to adjust the format according to the journal requirements.

Authors’ response: Thank you dear editors for your concern. We have putted the grant numbers in the funding information section. 

3. your Data Availability statement, you have not specified where the minimal data set underlying the results described in your manuscript can be found. PLOS defines a study's minimal data set as the underlying data used to reach the conclusions drawn in the manuscript and any additional data required to replicate the reported study findings in their entirety. All PLOS journals require that the minimal data set be made fully available. For more information about our data policy, please see

http://journals.plos.org/plosone/s/data-availability.

Upon re-submitting your revised manuscript, please upload your study’s minimal underlying data set as either Supporting Information files or to a stable, public repository and include the relevant URLs, DOIs, or accession numbers within your revised cover letter. For a list of acceptable repositories, please see

http://journals.plos.org/plosone/s/data-availability#loc-recommended-repositories. Any potentially identifying patient information must be fully anonymized.

Important: If there are ethical or legal restrictions to sharing your data publicly, please explain these restrictions in detail. Please see our guidelines for more information on what we consider unacceptable restrictions to publicly sharing data:

http://journals.plos.org/plosone/s/data-availability#loc-unacceptable-data-access-restrictions. Note that it is not acceptable for the authors to be the sole named individuals responsible for ensuring data access. We will update your Data Availability statement to reflect the information you provide in your cover letter

Authors’ response: Thank you dear editors for your concern. We have putted the appropriate data availability statement on the online submission.

4. Your ethics statement should only appear in the Methods section of your manuscript. If your ethics statement is written in any section besides the Methods, please move it to the Methods section and delete it from any other section. Please ensure that your ethics statement is included in your manuscript, as the ethics statement entered the online submission form will not be published alongside your manuscript.

Authors’ response: Thank you dear editors for your concern. We have putted it the method part according to your recommendation.

Response to reviewers #1

1. This is a cross sectional study to understand the determinants that affects QOL in patients with chronic diseases during COVID pandemic in Ethiopia. My comments are as follows:

The study design:

Although the authors mentioned in both discussion and conclusion that the quality of life was compromised during COVID-19, this conclusion cannot be drawn from the existing data. This is a cross sectional study - unless the current data can be compared to similar data prior to the COVID-19 outbreak, we cannot know if the QOL worsened during COVID-19.

Furthermore, the determinants that may predict QoL appeared to be like pre-COVID19 era. i.e., poorer QoL is found in people with disease complications. What's the new information then?

Authors’ response: Thank you dear reviewer for your concern. Of course, we didn’t make a before and after pandemic comparisons since we don’t have a data collected before the pandemic. The aim of this study was to inform the readers that the health-related quality of life of adults living with common chronic disease is compromised during the pandemic regardless of the pre-pandemic status. We used during the pandemic to refer the time or period of the study. The nobility of this study is quantifying the quality of life and its determinants in COVID 19 pandemic. It as identifies the direct indirect factors of quality of life.

Results:

2. The age of the participants are relatively young (they were all supposed to have chronic diseases). Why is that so? Most of the patients with chronic diseases are usually elderly. (Your IQR was 37-59).

Authors’ response: Thank you dear reviewer for your concern It is because of that nearly half of the study participants are adults with HIV/AIDS: it can occur more among sexually active individuals (younger age). Moreover, nearly 70% participants are within the age group of 40 or above with is plausible with the current understanding of in relation to chronic medical conditions. 

3. some of the results are difficult to read. for example, "age.... had direct positive effect on QOL". Does this mean younger or older age? The same goes with the other presentations.

Authors’ response: Thank you dear reviewer for your concern. We had thought that the term “direct and positive” could respond such kind of question. Now, revisions had been made accordingly to make it more clear for readers (See the last paragraph of the result section of the revised manuscript)

4. "kinds of medication.... had direct negative effective on QOL". How do you put different kinds of medications into a statistical model? Furthermore, what are the medications that are associated with low QOL?

Authors’ response: Thank you for your question. What we do mean by kind of medication is to refer how many types of medications a patient was taking. E.g., Participant x may say “they were taking glipizide, hydrocortisone, and prednisolone for the treatment of their condition. In this case, the data collectors recorded 3 in response to the how many kinds of medications a patient taking? Thus, we were not intended to assess the effect of a medicine on the quality of life of patient, instead we hypothesised and tested how increased number in kind of medications would interact with the quality of life of the patients. 

5. I will be very careful to use the word "Causal factors" because this is a cross-sectional study, many causal relationships cannot be established. For example, it is likely that people with good drug compliance have less problems with their diseases and thereby a better QOL. However, it may also be true that people with good QOL are less troubled by their problems and thereby remember to take their medications. This is one of the limitations that should be discussed.

Authors’ response: Thank you dear reviewer for your concern. Yes, casual inferences are not passible with a cross-sectional study. It is one of the limitations of such studies. We have acknowledged this in limitations of the study

Discussion:

6. most of the discussion were concentrated to explain the results. But some of the results may not need very detailed discussion. For example, it is well understood why presence of co-morbidities is linked to poorer QOL. There should be multiple reasons behind. Rather, the discussion can point out how we can use the results. Any policies that need to be changed according to your results? Any more research that needs to be done?

Authors response: Thank you for your comment. Revisions have been made accordingly (See the discussion section of revised manuscript)

7. The authors correctly pointed out that the study population is heterogeneous - it included multiple chronic diseases. Then, the authors can see if the results are the same when analysing only a subgroup of patients (e.g., only include patients with HIV or patients with hypertension). This will make your results more robust

Authors response: Thank you for your comment. We have tried to do a subgroup analysis. However, the sample in the groups subgroups was insufficient to run the subgroup analysis as structural equation modelling with a complex model requires a larger sample size. The model for the subgroups becomes unidentified model or not well converged.

8. The findings may apply to pre-COVID era. Can the data be compared to other studies in the pre-COVID era to show that there is indeed a worsening of environmental QOL, for example?

Authors response: Thank you for your question. To compare our finding with findings reported by studies conducted prior to COVID-19, there should be a study done with the same population and study area, at least. Unfortunately, to the best of the investigator’s knowledge, there are no studies done before the pandemic assessing QoL among these population in the study area. 

Minor:

9. There are multiple spelling and grammar mistakes. The manuscript may benefit from language editing.

Authors response: Thank you dear reviewer for your concern. English language review and editing were made by an English language professional, and revisions were made accordingly.

Response to reviewers #2

1. I don't have background on direct and indirect effects, the authors may want to clarify these to all readers. Overall, I understand the direct effect was determined via univariate model(s) whilst the indirect one was from a mixed multivariate model(s), then, why you must call it direct and indirect?

Authors Response: Thank you so much for your questions. 

The direct effect of one event on another can be defined and measured by holding constant all intermediate variables between the two or more outcomes. Indirect effects present a conceptual mediation effect because we cannot be isolated by holding certain variables constant. Our SEM examines how the quality of life (QOL) is affected through multiple mediators to predict a set of outcomes, such as physical, psychological, social, and environmental health. We want to see if the indirect effect through a set of the variables (e.g. sociodemographic variables  Var 1 Var 2 QoL) was a significant proportion of the main effect (sociodemographic variables  outcome).

2. Typo are found in the text, please proofread carefully

Authors Response: Thank you for your feedback, we performed detailed proofreading, checking spelling, grammar, sentence structure, and terminology with the help of language experts.

3. line 32: the range of mean score QOL seem to be large, author may consider reporting both mean and median

Authors Response: Thank you dear reviewer for your concern. We have tested its distribution and ii was not normal. As you know in the case of asymmetric distribution, the median with IQR is recommended.

4. line 161: the full term of CKD should be written before the abbreviation. Please check through all abbreviations.

Authors Response: Thank you dear reviewer for your concern. We made correction

5. line 171, 231: typo

Authors Response: Thank you dear reviewer for your concern, we mad correction as per your comment

6. line 205: reference of reliability thresholds should be cited properly.

line 262-266: also, many literatures have proved that QOL decrease by age (as their physicals health getting poorer), you may want to add some discussion on this phenomenon.

Authors Response: Thank you dear reviewer for your concern, revisions have been made. 

7. Figure 2 and 4: the author may want to add some notes to explain what are q1-q25, PHD, QOL, EHD, SRD, PSHD 

Figure 3: typo

Authors Response: Thank you dear reviewer for your concern, we include the details in the revised document.

Response to reviewers #3

1. Abstract and Introduction- Need to rephrase and construct for better sentences

Authors Response: Thank you dear reviewer for your concern, we tried to paraphrase it. 

2. Line 54-58 Need to check the sentences, the facts are contradicting with each other.

Authors Response: Thank you dear reviewer for your concern, revisions have been made. 

3. Line 96- suggest for ‘Facility based cross sectional study design’ to ‘A cross sectional study design’…

Authors Response: Thank you dear reviewer for your concern, revisions have been made. 

4. Methods- study setting

It is important to highlight during the study period, how many existing healthcare facilities are used for COVID-19, converted as hybrid healthcare facilities and healthcare facilities are not used for COVID-19. Not need to mention too detail about the burden of disease for COVID-19.

Can you explain about the semi-structured questionnaire used for your study?

Authors Response: Thank you dear reviewer for your concern, Semi-structured questionnaire means we have used both closed and open-ended questions. 

5. What is the proportion of face-to-face interview before you switch to the electronic form?

Authors Response: Thank you dear reviewer for your concern, we employed face-to-face interview with electronic data collection methods instead of paper. 

6. Thank you, dear reviewer, for your concern, why were the validity and reliability not conducted for SRQ-20 tool and Morisky Medication Adherence Scale (MMAS-8)?

Authors Response: Thank you dear reviewer for your concern, this tool is already widely used and validate for our country. 

7. The chronic diseases are self-reported or confirmed with the medical records?

Authors Response: Thank you dear reviewer for your concern, all diseases and outcomes are confirmed by appropriate physicians and are already recorded in the patients’ medical chart. 

8. There are few typos error

Authors Response: Thank you dear reviewer for your concern, revisions have been made. 

9. How do you overcome information and recall bias in this study?

Authors Response: it is not totally free from biases, but We tried to probe the patients by raising different scenario. We also crosscheck with their medical records 

10. Should separate the categories able to read and write and the education level and re-analyze the data.

Authors Response: Thank you for your concerns. First, we tried to fit a model that includes with educational level. However, the major limitation of SEM is that the determinants should be either continuous variable or binary if it is categorical.

11. Can the patient the differentiate the symptoms of the disease and complications of the disease?

Authors Response: Thank you dear reviewer for your concern. We conduct the chart review and take from their chart since all diseases and outcomes which are confirmed by physicians are already recorded in the patients’ medical chart. 

12. How did you ask the common mental disorders for this study? (This is a diagnosis)

Authors Response: Thank you dear reviewer for your concern. We used self-reported common mental disorders SRQ-20 tool and finally coding was conducted based on the guideline of the tool.

13. What was measured for ‘Kind of medication’?

 Authors Response: Thank you dear reviewer for your concern. We asked them how many kinds of drugs they took, and its unit was in numbers.

---

## [Decision Letter · Decision Letter 1]

15 Aug 2022

PONE-D-21-23936R1Quality of life among patients with common chronic disease during COVID-19 Pandemic in Northwest Ethiopia; A Structural Equation ModelingPLOS ONE

Dear Dr. Alamneh,

Thank you for submitting your manuscript to PLOS ONE. After careful consideration, we feel that it has merit but does not fully meet PLOS ONE’s publication criteria as it currently stands. Therefore, we invite you to submit a revised version of the manuscript that addresses the points raised during the review process.

We look forward to receiving your revised manuscript.

Kind regards,

Filipe Prazeres, MD, MSc, Ph.D.

Academic Editor

PLOS ONE

Journal Requirements:

Reviewers' comments:

Reviewer's Responses to Questions

**Comments to the Author**

1. If the authors have adequately addressed your comments raised in a previous round of review and you feel that this manuscript is now acceptable for publication, you may indicate that here to bypass the “Comments to the Author” section, enter your conflict of interest statement in the “Confidential to Editor” section, and submit your "Accept" recommendation.

Reviewer #3: All comments have been addressed

2. Is the manuscript technically sound, and do the data support the conclusions?

Reviewer #3: Yes

3. Has the statistical analysis been performed appropriately and rigorously? 

Reviewer #3: No

4. Have the authors made all data underlying the findings in their manuscript fully available?

Reviewer #3: Yes

5. Is the manuscript presented in an intelligible fashion and written in standard English?

Reviewer #3: Yes

6. Review Comments to the Author

Reviewer #3: Refer to the attachment file.

1. Please put the Cronbach Alpha value for SRQ-20 tool and Morisky Medication Adherence Scale (MMAS-8), and cited the papers for tools validation conducted in Ethiopia(as references).

2. For descriptive analysis (table 1), ‘Unable to read and write’ and ‘Able to read and write’ should be put under new variable e.g ‘Literacy status’. The 2 categories mentioned above are not compatible with the existing categories in the variable ‘education status’

For SEM analysis, you need to decide how to make the 3 categories for education status into binary. (same with marital status that have more than 2 categories)

3.The words ‘common mental disorder’ in the narrative of the results, tables, discussion and conclusion are not appropriate, because SRQ-20 tool is a screening tool for depression, anxiety symptoms and psychosomatic complaints.

Please change ‘common mental disorder’ with ‘mental health problem’.

4.The use of ‘Kind of medication’ is not clear, unless you define it in the operational definition.

It is suggested the variable for ‘Kind of medication’ is change to ‘how many medication’.

7. PLOS authors have the option to publish the peer review history of their article (what does this mean?). If published, this will include your full peer review and any attached files.

Reviewer #3: No

---

## [Author Response · Author response to Decision Letter 1]

24 Aug 2022

August 2022

Rebuttal letter

Submission ID: PONE-D-21-23936R1

Title: Quality of life among patients with common chronic disease during COVID-19 Pandemic in Northwest Ethiopia; A Structural Equation Modelling

PLOS ONE

Tadesse Awoke Ayele, Habtewold Shibru, Malede Mequanent Sisay, Tesfahun Melese, Melkitu Fentie, Telake Azale, Tariku Belachew, Kegnie Shitu, and Tesfa Sewunet Alamneh*

Dear Editor and reviewer, 

We would like to thank for your consideration and suggestion for the betterment our manuscript and make it more informative. We tried to amend the format of the manuscript according to the journal guidelines and address the questions raised by reviewer on the manuscript. Our point-by-point responses for each comment and questions are described in detail on the following pages. Further, the details of changes were shown by track changes in the supplementary document attached. 

Response to Reviewer 

1. Please put the Cronbach Alpha value for SRQ-20 tool and Morisky Medication Adherence Scale (MMAS-8) and cited the papers for tools validation conducted in Ethiopia (as references).

Author’s response: thank you dear reviewer and we appreciate your comment. We have included it.

2. For descriptive analysis (table 1), ‘Unable to read and write’ and ‘Able to read and write’ should be put under new variable e.g., ‘Literacy status’. The 2 categories mentioned above are not compatible with the existing categories in the variable ‘education status’

Author’s response: thank you dear reviewer and we appreciate your comment. We have changed to literacy status.

3. The words ‘common mental disorder’ in the narrative of the results, tables, discussion, and conclusion are not appropriate, because SRQ-20 tool is a screening tool for depression, anxiety symptoms and psychosomatic complaints.

Please change ‘common mental disorder’ with ‘mental health problem’.

Author’s response: thank you dear reviewer and we appreciate your comment. We have updated it. 

4. The use of ‘Kind of medication’ is not clear unless you define it in the operational definition. It is suggested the variable for ‘Kind of medication’ is change to ‘how many medication’.

Author’s response: thank you dear reviewer and we appreciate your comment. It was supposed to indicate how many medications were taken. As you mentioned, it is ambiguous, and we changed it to kind of medication according to make it clear.

---

## [Decision Letter · Decision Letter 2]

19 Sep 2022

PONE-D-21-23936R2Quality of life among patients with common chronic disease during COVID-19 Pandemic in Northwest Ethiopia: A Structural Equation ModellingPLOS ONE

Dear Dr. Alamneh,

Thank you for submitting your manuscript to PLOS ONE. After careful consideration, we feel that it has merit but does not fully meet PLOS ONE’s publication criteria as it currently stands. Therefore, we invite you to submit a revised version of the manuscript that addresses the points raised during the review process.

We look forward to receiving your revised manuscript.

Kind regards,

Filipe Prazeres, MD, MSc, Ph.D.

Academic Editor

PLOS ONE

Journal Requirements:

Reviewers' comments:

Reviewer's Responses to Questions

**Comments to the Author**

1. If the authors have adequately addressed your comments raised in a previous round of review and you feel that this manuscript is now acceptable for publication, you may indicate that here to bypass the “Comments to the Author” section, enter your conflict of interest statement in the “Confidential to Editor” section, and submit your "Accept" recommendation.

Reviewer #3: All comments have been addressed

2. Is the manuscript technically sound, and do the data support the conclusions?

Reviewer #3: Yes

3. Has the statistical analysis been performed appropriately and rigorously? 

Reviewer #3: No

4. Have the authors made all data underlying the findings in their manuscript fully available?

Reviewer #3: Yes

5. Is the manuscript presented in an intelligible fashion and written in standard English?

Reviewer #3: Yes

6. Review Comments to the Author

Reviewer #3: 1. For descriptive analysis (table 1), you only changed the ‘education status’ to ‘Literacy status’ and the categories remained the same and also their frequency and percentage.

(i) for the new variable ‘Literacy status’, the 2 categories under it are ‘Unable to read and write’ and ‘Able to read and write’. The total percentage of these 2 categories is 100%. Do not include the categories under variable ‘education status’ in this variable ‘ literacy status’.

(ii) Please maintain the variable ‘education status’. The 3 categories in this variable are Primary education, Secondary education and Diploma. Please ensure the total percentage of these 3 categories is 100%.

2. Please address my comment in the previous revision:

For SEM analysis, you need to decide how to make the 3 categories for education status into binary. (same with marital status that have more than 2 categories)

7. PLOS authors have the option to publish the peer review history of their article (what does this mean?). If published, this will include your full peer review and any attached files.

Reviewer #3: No

---

## [Author Response · Author response to Decision Letter 2]

17 Oct 2022

October 2022

Rebuttal letter

Submission ID: PONE-D-21-23936R1

Title: Quality of life among patients with common chronic disease during COVID-19 Pandemic in Northwest Ethiopia; A Structural Equation Modelling

PLOS ONE

Tadesse Awoke Ayele, Habtewold Shibru, Malede Mequanent Sisay, Tesfahun Melese, Melkitu Fentie, Telake Azale, Tariku Belachew, Kegnie Shitu, and Tesfa Sewunet Alamneh*

Dear Editor and reviewer, 

We would like to thank for your consideration and suggestion for the betterment our manuscript and make it more informative. We undergone editing of the manuscript to address the questions raised by reviewer. Our point-by-point responses for both comments are described in detail on the following pages. Further, the details of changes were shown by track changes in the supplementary document attached. 

Response to Reviewer 

1. For descriptive analysis (table 1), you only changed the ‘education status’ to ‘Literacy

status’ and the categories remained the same and their frequency and percentage.

(i) for the new variable ‘Literacy status’, the 2 categories under it are ‘Unable to read and

write’ and ‘Able to read and write’. The total percentage of these 2 categories is 100%. Do

not include the categories under variable ‘education status’ in this variable ‘literacy status’.

(ii) Please maintain the variable ‘education status’. The 3 categories in this variable are

Primary education, Secondary education, and Diploma. Please ensure the total percentage of these 3 categories are 100%.

Author’s response: thank you dear reviewer and we appreciate your effort for the betterment of our work. We have tried to address your comment by including literacy status and educational level. As you mentioned it the literacy variable has 2 categories who can read and write and unable to read and write. Able to read and write individual were further classified to the three categories that you strike it. The frequency and detail editing are found in the revised version.

2. Please address my comment in the previous revision:

For SEM analysis, you need to decide how to make the 3 categories for education status into

binary. (Same with marital status that have more than 2 categories).

Author’s response: thank you dear reviewer and we appreciate your comment. After you suggestion we used to literacy status (which is binary in nature) instead of educational level after assuming all educated individual can read and write. We assumed that separated, widowed, and divorced individuals were married at certain time but it might cause loss of information since there might be heterogeneity among married, separated, widowed, and divorced individuals. Due this, we include the drawbacks of SEM which only consider either continuous or binary variable in the measurement model as a limitation in our work.

---

## [Editor Report · Decision Letter 3]

21 Nov 2022

Quality of life among patients with common chronic disease during COVID-19 Pandemic in Northwest Ethiopia: A Structural Equation Modelling

PONE-D-21-23936R3

Dear Dr. Alamneh,

We’re pleased to inform you that your manuscript has been judged scientifically suitable for publication and will be formally accepted for publication once it meets all outstanding technical requirements.

Kind regards,

Filipe Prazeres, MD, MSc, Ph.D.

Academic Editor

PLOS ONE
---

## [Editor Report · Acceptance letter]

25 Nov 2022

PONE-D-21-23936R3 

Quality of life among patients with the common chronic disease during COVID-19 Pandemic in Northwest Ethiopia: A Structural Equation Modelling 

Dear Dr. Alamneh:

I'm pleased to inform you that your manuscript has been deemed suitable for publication in PLOS ONE. Congratulations! Your manuscript is now with our production department. 

Kind regards, 

on behalf of

Prof. Filipe Prazeres 

Academic Editor

PLOS ONE